# Current Advancements in Noninvasive Profiling of the Embryo Culture Media Secretome

**DOI:** 10.3390/ijms22052513

**Published:** 2021-03-03

**Authors:** Raminta Zmuidinaite, Fady I. Sharara, Ray K. Iles

**Affiliations:** 1MAP Sciences Ltd., The iLab, Stannard Way, Priory Business Park, Bedford MK44 3RZ, UK; Raminta.Zmuidinaite@mapsciences.com; 2Virginia Center for Reproductive Medicine, Reston, VA 20190, USA; fsharara@vcrmed.com; 3NISAD (Lund), Medicon Village, SE-223 81 Lund, Sweden

**Keywords:** IVF, embryo screening, noninvasive, proteomics, metabolomics

## Abstract

There have been over 8 million babies born through in vitro fertilization (IVF) and this number continues to grow. There is a global trend to perform elective single embryo transfers, avoiding risks associated with multiple pregnancies. It is therefore important to understand where current research of noninvasive testing for embryos stands, and what are the most promising techniques currently used. Furthermore, it is important to identify the potential to translate research and development into clinically applicable methods that ultimately improve live birth and reduce time to pregnancy. The current focus in the field of human reproductive medicine is to develop a more rapid, quantitative, and noninvasive test. Some of the most promising fields of research for noninvasive assays comprise cell-free DNA analysis, microscopy techniques coupled with artificial intelligence (AI) and omics analysis of the spent blastocyst media. High-throughput proteomics and metabolomics technologies are valuable tools for noninvasive embryo analysis. The biggest advantages of such technology are that it can differentiate between the embryos that appear morphologically identical and has the potential to identify the ploidy status noninvasively prior to transfer in a fresh cycle or before vitrification for a later frozen embryo transfer.

## 1. Introduction

Since the first in vitro fertilization (IVF) in 1978, the number of babies born with the help of IVF has exceeded 8 million and continues to grow year by year. With over 40 years of research in the field, there were significant improvements made. However, the efficiency of embryo implantation remains low, ranging from 4% to 40% [1]. Therefore, increasing the effectiveness of the procedure is the desired outcome and has been the focus of researchers in the field of reproductive biology and medicine. One way to increase the chance of live birth and reduce the time to pregnancy is to screen the embryo prior to transfer. This encompasses embryo profiling for the selection of the embryo most likely to implant and leads to an intrauterine pregnancy, identification of aneuploidies and embryo development competence. To date, numerous approaches and embryo scoring tools have been proposed. Needless to say, the IVF procedure is financially costly and stressful physically and psychologically for the families involved. These embryo screening methods aim to increase the chances of live births, therefore decreasing the financial burden and reducing the anxiety, depression and stress especially associated with failed IVF cycles.

In the past decades, the embryo assessment gold standard methods did not change considerably and remain the microscopy of static developmental time points performed by highly trained embryologists [2]. There have been numerous publications discussing the limitations of the morphology assessment method. Most of them were due to the subjectivity of the embryologist, suboptimal culture conditions, and poor association with embryo implantation potential. This gave rise to numerous improvements, including trying to eliminate the bias and standardize the media culture environment by employing artificial intelligence (AI) and time-lapse imaging (TLI). Some of the key findings will be reviewed in Section 2.2.

Preimplantation genetic testing (PGT) is designed to test the embryos for inherited chromosomal abnormalities that could impair embryo implantation, development or the health of the baby. The current gold standard methods require a biopsy to obtain genetic material from the developing embryo. Potential risk of compromising embryo development, long-term consequences [3,4] and associated high costs with the biopsy procedure have encouraged scientists to investigate options for a noninvasive PGT (niPGT), correlating genetic material found in blastocoele fluid (BF) and spent blastocyst media (SBM) to the one in trophectoderm (TE) following the biopsy. A recent review was published on the current advancements in the field niPGT testing for the identification of embryo ploidy. It suggests that both BF and SBM could be sufficient and more efficient means for chromosomal abnormality detection. Yet, there are still many challenges to overcome when compared to the conventional biopsy method [5]. However, although aneuploidy alone may account for a significant number of failed implantations, embryo maturity and non-gross chromosomal events during early embryogenesis are probably the greater contributors to failure. Thus, instead of correct identification of aneuploidy within trophectoderm cells, the end measure of an IVF embryo selection test should be increased live birth rates, all be it incorporating ploidy as a measure [6,7,8].

Given the cons of the current gold standard methods for embryo profiling, scientists have been investing in the development of noninvasive techniques. Implementing a noninvasive approach to predict embryo outcomes would be beneficial in many ways, reducing the chance of impairing the embryo, its environment and, therefore, development potential. It would also reduce costs, first by eliminating the need for the biopsy and reducing the number of cycles by increased efficacy of such tests. It is therefore important to understand where current research of noninvasive testing for embryos stands and what are the most promising techniques currently used. Furthermore, this understanding is key to identify the potential to translate research and development into clinically applicable methods that ultimately improve live birth and reduce time to pregnancy.

Noninvasive testing opens new perspectives in both the determination of the ploidy status and the selection of the embryo with the most potential for the live birth outcome. The aim of this review is to summarize the current advancements in profiling of the embryo by the novel, noninvasive methods, focusing on the spent media profiling.

## 2. Noninvasive Testing, Current Focus

The main focus of the studies evaluating embryo potential is to improve the prioritization of the most viable embryo for an elective single embryo transfer (eSET). This is the most effective way to avoid multiple gestations that are associated with adverse medical conditions for both mother and baby [9] and to achieve a healthy singleton infant, the most desired outcome in assisted reproduction. There are many techniques that have been applied for the noninvasive analysis of embryo quality, all at different stages, from proof-of-concept studies showing great potential to those already used in IVF clinics (Table 1). For the differentiation of embryos with the most potential, there are three main currently trending research areas. One is the time-lapse microscopy empowered by AI-based computational methods; second is the analysis of cell-free DNA released by the embryo into the spent blastocyst medium (SBM) to determine the status of embryo ploidy; and lastly, the correlation of metabolomic/proteomic markers, present in the SBM, to embryo viability.

While developing any innovative technique, it is critical to ask whether the tested embryo is clinically transferable. Furthermore, is the test user-friendly and can it be readily deployed in the laboratory, subsequently improving IVF outcomes? All of these factors are considered by the directors of the IVF clinics that would be eventual decision-makers for the adaptation of the technique. For example, as indicated by the national survey, based on 294 IVF clinics in the United States, the majority (60.3%) do not own the time-lapse monitoring system and have no plans to get one [10]. A French National Survey found that even when the majority of non-users (56.8%) and users (93.1%) of TLI agree that TLI is superior to standard morphology, barely 30.2% and 51.7%, respectively, think that TLI will soon become the gold standard for embryo quality assessment [11]. In order to bring the change in the laboratory, the methodology should be easy to adapt, user-friendly, and prevent tinkering with embryos, potentially compromising implantation potential.

The current focus in the field of human reproductive medicine is to develop a more rapid, quantitative, and noninvasive test. Regardless of the technique used, the final goal of the embryo testing is a healthy baby delivered at term and noninvasive techniques are offering the solution without the requirement of tampering with the embryo cells. Some of the most promising fields of research comprise cell-free DNA analysis, microscopy techniques coupled with AI and omics analysis of the SBM.

### 2.1. Cell-Free DNA Analysis

Introduced in 2011, the cell-free DNA method in reproductive medicine was applied for the acquisition of circulating fetal DNA from the mother’s blood plasma in early gestation [36]. This concept was later adopted for the screening of embryos prior to implantation, enabling a noninvasive sampling. Studies are suggesting consistent results in comparison to invasive TE biopsy with a fairly high concordance for ploidy results [37,38,39]. However, in general, the reported concordance rates between SBM and TE biopsy are heterogenous, varying from 15.4% to 100% [15]. Cell-free DNA is primarily employed for the identification of embryo ploidy, but the technique was also used to correlate other outcomes such as pregnancy rate and embryo quality. Quantitative analysis of the ratio between genomic (gDNA) and mitochondrial (mtDNA) DNA was shown to be significantly (*p* < 0.0001) higher in the embryos with high development potential and successful implantation outcome (*p* = 0.045) [40].

An advantage of cell-free DNA analysis from SBM, compared to TE biopsy, is that it may lower the false positives and false negatives. SBM is a more representative sample of the whole embryo and bypasses the issues surrounding human embryo mosaicism [5]. Furthermore, the technically challenging and invasive TE biopsy does not always represent the inner cell mass (ICM) [41]. Moreover, it is debated that the DNA found in SBM could contain genetic contamination of maternal cell origin [42]. The recent systemic review on the cell-free DNA for the human embryo ploidy assessment excluded several disadvantages. They concluded that even though genetic material was successfully detected and amplified, reliability due to arising discrepancies is still debated [15]. The main discrepancy sources identified were a low amount of DNA, the varying selection of reference sources for concordance studies, or potential contamination with exogenous DNA.

### 2.2. Microscopy Techniques for Noninvasive Embryo Screening

The current gold standard for embryo assessment is the static observation at development times using microscopy techniques, which essentially are decision trees. The system has been developed through years of observation and become more sophisticated over time [43]. The rate of embryo development was one of the key indicators, and additional discrete stages of development related to implantation potential were later incorporated into decision analysis. Nowadays, such systems are increasingly being enhanced by advanced computational techniques, employing computer vision static image processing and developing AI-based models for embryo viability prediction [18]. The advantage of these models is that they are not biased and do not suffer from low concordance rates between qualified embryologists.

While the current gold standard microscopy relies heavily on an expert embryologist, the introduction of AI models can help reduce the bias. However, these AI models still depend on a user-defined input, such as morphological characteristics. Nevertheless, morphological grading reached the next level with the introduction of time-lapse microscopy almost a decade ago. An order of magnitude of embryo kinetics and morphology data became available and were shown to produce a superior outcome in comparison to conventional screening, when measuring implantation rates [22,44,45]. It has been also demonstrated that such AI-based models could outperform an expert embryologist across multiple clinics, achieving an Area Under the Curve (AUC) of 0.82 ± 0.07 for a computational prediction model and an AUC of 0.58 ± 0.04 for an expert embryologist panel, when measuring implantation rates retrospectively [46]. Furthermore, time-lapse monitoring was applied in the determination of the ploidy status, with six selected kinetic parameters found to differentiate between normal and abnormal embryos [47]. However, another group found no correlation between sixteen common morphokinetic markers of in vitro embryo development and ploidy status [48]. Thus, assessment of embryo ploidy by time-lapse requires further research.

### 2.3. Spent Blastocyst Media Analysis

Viable embryo selection is predominantly based on the morphological evaluation, including morphokinetics, which is often carried out subjectively. This assessment of morphology is not necessarily an absolute link to the health of the embryo and a resulting child. The analysis of SBM offers an additional layer of information beyond that of embryo appearance. Furthermore, unlike morphological assessment, SBM analysis goes beyond embryo quality evaluation and prediction of implantation. It has been hypothesized that a preimplantation metabolic profile of the environment of the embryo could be indicative of the future offspring health, via the influence of the embryo’s epigenetic state [49]. This could offer information beyond implantation and childbirth potential.

SBM surrounds the embryo during the developmental preimplantation phase. Both the utilization of nutrients present in the media and secretion of metabolites by the embryo can play a part in providing information about the embryo’s status. Figure 1 is a schematic of the materials in SBM that potentially provide insights on what decisions about embryo quality and ploidy status can be made. These samples enable research methodologies for specific biomarker detection, genomics, transcriptomics, metabolomics, and proteomics.

For specific marker detection, many studies that analyzed the presence of the soluble human leukocyte antigen G (sHLA-G) concluded that the standardized sHLA-G assay has the potential to identify the most competent embryos for implantation. An increased secretion of this biomarker was found to be associated with a successful pregnancy [50,51,52]. The secretion of sCD146, as measured by adapted commercial enzyme-linked immunosorbent assay (ELISA), was shown to be significantly (*p* = 0.00624) associated with a lower implantation potential [19]. A multi-center study provides significant evidence that the morphological scoring system is still the best strategy for the selection of embryos, but that sHLA-G might be considered as a second parameter in combination for improved outcomes [51]. These biochemical assays, with easily obtainable commercial kits, can be particularly useful for an additional layer of information if there is more than one embryo with the same morphology for a selective single ET. Furthermore, SBM enables the study of cell-free DNA, as discussed in Section 2.1, and transcriptomics. Borges et al. identified an miRNA (*miR-142-3p*) to be significantly correlated with implantation failure; thus, embryos producing this miRNA could be excluded from implantation [53]. Overall, transcriptomics gives insight beyond genetics, informing about epigenetic modifications, and could be a valuable tool [54].

Finally, unlike immunological assays or genetic/transcriptomic screening, targeting specific genes/transcripts, metabolomic and proteomic assays have the main advantage that they are not limited to specific biomarkers. Moreover, recent improvements in the hardware mean that the tests can be performed immediately prior to transfer, with high accuracy and increased effectiveness. The subjective data interpretation is eliminated by the use of bioinformatics pipelines, analyzing high-dimensional data. It is, therefore, reasonable to presume that SCM could be an accurate source of the medium, enabling an eSET with the highest potential for the live birth outcome.

## 3. Proteomics and Metabolomics of Spent Blastocyst Media

High-throughput proteomics and metabolomics technologies are valuable tools to study the molecular components of biological systems. The biggest advantage of such technology is that it can differentiate between embryos that appear morphologically identical. It also has the potential to identify the ploidy status noninvasively prior to transfer or vitrification for a later frozen cycle. However, one of the issues with metabolomic and proteomic mass spectrometric profiling is the scale and amount of generated data from a single sample. It requires cutting-edge data processing pipelines to effectively extract the meaning from the generated data and to build meaningful algorithms optimized for the desired outcome [55].

Another important factor to consider when analyzing SBM is environmental factors such as temperature, humidity, and air quality. They have been shown to affect epigenetics and, subsequently, embryo morphology, developmental kinetics, physiology and metabolism [56]. Other researchers looked at the differences in the commercial culture media and found no significant difference (*p* = 0.521) as measured by singleton birth weight [57]. Strict procedures and protocols employed by both the manufacturers of culture media and IVF clinics play an important role in standardizing the approach, reducing the impact of external factors. Using continuous commercially made single media may be advantageous overall, reducing variation and thus making proteomic and metabolomic techniques for SBM more readily adaptable [58].

### 3.1. Current Techniques

The focus of the development of novel applications is high accuracy, sufficient to enable an eSET. Both proteomics and metabolomics are complementary platforms and can be applied using a high-throughput methodology. Table 1 summarizes the techniques that are currently being studied for the embryo screening. One important consideration for the use of SBM as a means for analysis is that developing several embryos in a single droplet would be not a viable option. The discriminative power of selecting a single embryo would be diminished if multiple embryos are developing in a single droplet. This, in turn, could require more incubators, time, and space for embryo development in individual media droplets. Nevertheless, given the benefits of rapid and low-cost screening, it could be a worthwhile added cost. Proteomics, specifically mass spectrometry, is a rapidly developing technology and could be successfully applied for the analysis of SBM. Technology such as Matrix-Assisted Laser Desorption Ionization Time of Flight Mass Spectrometry (MALDI-ToF MS) has the same limitations as any other technique; however, when used in accordance with strict Standard Operating Procedures (SOPs) it is highly reproducible [59].

Larger scale meta-analyses or systemic reviews are lacking. It is challenging to compare multiple studies. Mainly, this is due to the general lack of standardized outcomes, study designs and the variability of proteomics and metabolomics techniques used. Outcome variables selected for the study vary from embryo development potential and implantation potential with a positive human chorionic gonadotropin (hCG) test or with the fetal heart rate to live birth rates. However, one meta-analysis of four NIR studies found no evidence that live birth rates were improved [60]. 

### 3.2. Biomarkers of Interest

In the mass spectrometric analysis, a large number of biomarkers are identified after a single run of the sample. Some optimization and standardization are required as there could be massive variation in the spectral profile, depending on the chemistry used in the sample pre-processing and acquisition steps. There are two approaches pursued, one being a targeted identification of specific biomarkers and the other analysis of the correlation of the spectral pattern as a whole to the outcome of interest.

A number of techniques, particularly in metabolomics, are applied to analyze and identify specific biomarkers, their ratios and their concentrations in SBM. Section Spent Media Analysis of Table 1 lists these methods. With roughly 3000 potential metabolites, there is a space for possibilities of different markers in the analysis of embryo viability as well as genetic status. LC-MS/GS-MS and NMR were successfully employed to differentiate between trisomy/monosomy 21 and euploid SBM. Two metabolites, caproate and androsterone sulphate, were identified [61]. Twelve metabolites were identified and analyzed by 1H-NMR (proton-NMR), identifying that the increase in formate to glycine ratio and the decrease in citrate to alanine ratio was indicative of intrauterine pregnancy [28]. Metabolomic profiling by Raman spectroscopy identified sodium pyruvate and phenylalanine levels to be associated with embryo implantation potential [62]. The principal component and discriminative analysis were applied to amino acid spectral data obtained by HPLC. Authors were able to predict the amino acid fingerprint and embryo implantation potential with high accuracy (90.4%) [21]. Differential expression of human chorionic gonadotropin (hCG) isoforms [63] and interleukin (IL) 6, stem cell factor (SCF) and interferon (IFN) α2 [64] were identified to be suggestive of embryo success. Another research group identified 18 exclusively expressed proteins in the positive implantation group and 11 in the negative embryo implantation group [65]. These studies are promising; however, most are suggestive and use a small cohort (average SBM sample number = 71), thus requiring further validation. The systemic review by Bracewell-Milnes et al., on metabolomic biomarkers in reproductive medicine, could be useful for further comprehensive reading [66].

### 3.3. High-Throughput SBM Analysis

The SBM is a rich source of biological material altered directly by the embryo, thus can be reflective of its quality and health. Both the secreted proteins into the media as well as the uptake of the proteins already present in SBM can be valuable biomarkers of embryo development [49]. High-throughput proteomic tests utilizing these biomarkers could enable the eSET with the highest potential embryo. It is argued that it is necessary to identify the putative biomarkers and, following clinical validation, develop specific antibody-based methods [67]. This step could help validate the technology and increase its acceptance in a clinical setting. However, the ultimate goal is the improvement of the live birth numbers. Therefore, mass spectral fingerprinting, without identification of the underlying protein biomarkers, could be a useful and rapid means for embryo screening. The complexity of embryo development is appreciated more when profiling is not limited to a single biomarker. It has been shown that mass spectral fingerprinting could be useful in an array of applications, from cancer screening to microbiology and prenatal screening, detecting biochemical changes in biofluids [55]. The same principle is to be applied for the high-throughput SBM analysis. The analysis of thousands of proteins is performed in an unbiased and systemic manner. MALDI-ToF MS is a relatively straightforward technique in terms of sample pre-processing. This proteomic method was applied in the discrimination of PGT-A tested embryos, characterizing 12 unique spectral regions for euploid and 17 for aneuploid patterns [25]. Similarly, the selection of the best embryo for transfer has shown great results with a positive predictive value for ongoing pregnancy of 82.9% [68].

Advances in the hardware of mass spectroscopy (MS) over the recent decades make it an ideal tool for high-throughput analysis, which is fast, cost-effective, and sensitive. Furthermore, current, state-of-the-art instruments on the market are low-profile and bench-top, making them appropriate for limited space in IVF clinics. The cost comparison of MS with current PGT-A methods is tenfold less [25], in hundreds instead of thousands of dollars for a patient. The maintenance of mass spectrometers is usually outsourced to the network of suppliers. Furthermore, since the SBM samples are volume-limited, MS offers an additional advantage by requiring as low as 1 µL of the sample volume. Figure 2 shows a pipeline of a high-throughput embryo analysis from a petri dish to a computational score outcome for the individual embryos. To further read on bioinformatic pipeline workflow development for high-throughput proteomic applications, refer to Pais et al. [69].

## 4. Future Perspectives

There is a diverse range of methods being developed for the profiling of embryos for IVF treatments in order to increase the number of live births per cycle. However, many of the studies reported are retrospective, pilot studies, with a limited number of SBM samples. There is a lack of prospective randomized studies that would demonstrate the increase in the implantation and live birth rates. Furthermore, not all pilot or proof-of-concept studies take into account the clinical applicability of the technology, limiting the potential to translate research and development into clinical practice. It is important to weigh the applicability of the contemporary methods in an IVF clinical setting. It could be relatively easy to implement the new low-profile, bench-top MALDI-TOF MS models or time-lapse microscopy combined with AI. Yet, approaches using HPLC or LC-MS/GS-MS could be less feasible for routine embryo screening.

These applications could be an alternative method to increase the screening for aneuploidies in cases where invasive PGS is not permitted or not desired. It could expand the availability of screening in those cases where it is not available due to the high costs of current technologies. It could be a major shift in the paradigm of how IVF embryos are screened prior to the implantation, enabling a safer and more efficient alternative. Furthermore, besides reducing the time to pregnancy and risk of an aneuploid pregnancy, these methodologies could be beneficial from the health economic perspective. Using eSET could help reduce multiple pregnancies and therefore the associated patient, healthcare and societal costs [70].

Artificial intelligence (AI) is playing a major role in improving and assisting in many of the methods in both microscopy and spent media analysis fields. There are numerous studies emerging that support the application of various technologies by developing automated annotation software for morphokinetics analysis [25,63,68,71,72,73]. These new automated and integrative computational approaches open doors to less biased, rapid, robust and ultra-fast results from the moment data are available. Combined with high-throughput proteomics assays, it could truly lead to a paradigm shift in how an ever-increasing number of embryos are screened.

## 5. Conclusions

The recent research in the field of reproductive medicine has shown that the trend for noninvasive methods for simple and direct analysis is necessary. The aim is to ensure the accurate selection of the best embryo for transfer, ensuring the minimization of time to pregnancy. With current developments in both high-throughput, clinically applicable hardware and bioinformatic tools, this goal is getting closer to being reached. Nevertheless, there is still some work needed to translate the pilot studies into the clinical arena.

## Figures and Tables

**Figure 1 ijms-22-02513-f001:**
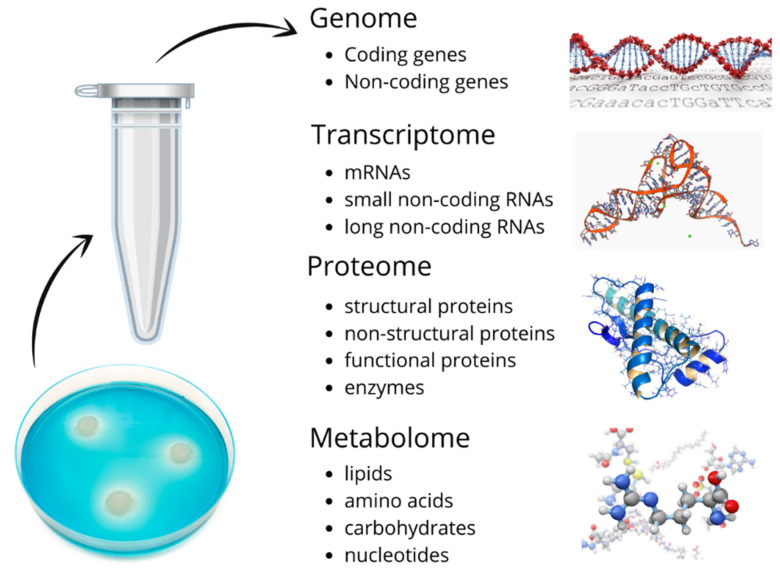
Schematic of materials found in spent blastocyst medium (SBM) that pose the diagnostic potential for embryo quality and ploidy status.

**Figure 2 ijms-22-02513-f002:**
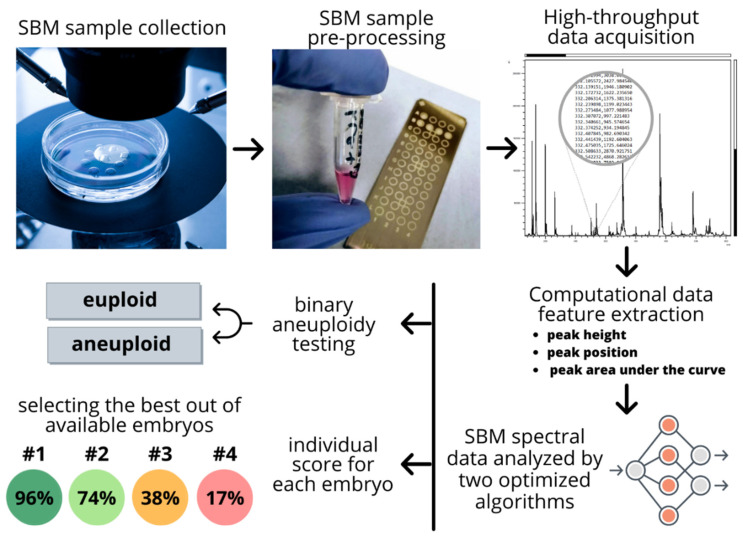
Pipeline of a high-throughput embryo analysis from a petri dish to a computational score outcome for the individual embryos. The pipeline goes from a non-invasive spent blastocyst medium (SBM) sample collection to the obtainment of final scoring that facilitated selection of the best available embryo for implantation.

**Table 1 ijms-22-02513-t001:** Summary of noninvasive assays for embryo quality assessment with a section of recent references.

Microscopies	References	Spent Media Assays	References
Auto-fluorescence	[12]	Array Comparative Genomic Hybridization (aCGH)	[13]
Fluorescence-lifetime imaging (FLIM)	[14]	Cell free DNA genetic testing (PCR)	[15]
Hyperspectral imaging	[16]	Electrochemical Impedance Spectroscopy (EIS)	[17]
Light microscopy	[18]	Enzyme-linked immunosorbent assay (ELISA)	[19]
Polarization microscopy	[20]	High performance liquid chromatography (HPLC)	[21]
Time-lapse imaging	[22]	Liquid chromatography/gas chromatography coupled with LC-MS/GS-MS	[23,24]
		Matrix-Assisted Laser Desorption Ionization Time of Flight Mass Spectroscopy (MALDI-ToF MS)	[25]
		Microarray	[26]
		Microfluidics	[27]
		Nuclear Magnetic Resonance spectroscopy (NMR)	[28]
		Proximity Extension Assay (PEA)	[29]
		Respirometry	[30]
		Thermochemiluminescence (TCL)	[31]
		Ultramicrofluorescence (UMF)	[32]
		Vibrational spectroscopies (FTIR/NIR/RS)	[33,34,35]

## Data Availability

Restrictions apply to the availability of these data.

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
