# Peer review of "Current Advancements in Noninvasive Profiling of the Embryo Culture Media Secretome"

_ijms, 2021, doi:10.3390/ijms22052513_

Round 1

Reviewer 1 Report

Authors review in a comprehensive manner the current noninvasive techniques to assess embryo quality in the field of human reproductive medicine and its application at the clinical level. Authors conclude that the most promising fields of research comprise cell-free DNA analysis, microscopy techniques coupled with artificial intelligence (AI) and omics analysis of the spent blastocyst medium (SBM). However, this reviewer doubts about its full implantation at a clinical level within a short period of time.

Some points to be expanded and corrected:

  1. As for the reported heterogenous concordance rates between SBM and trophectoderm biopsy authors should discuss about the potential impact, if any, of different culture media and protocols among clinics. Also, for SBM technique, the different commercial culture medium, whose composition is unknown for the clinicians, can mask, increase or decrease the embryo secretion of specific biomarkers and metabolites. Same applies on the effect of culture media employed on morphological characteristics and embryokinetics. Some suggested papers to introduce this concept might be Dyrlund et al. 2014 (Human Reproduction Nov;29(11):2421-30. doi: 10.1093/humrep/deu220), De Vos et al. 2015 (Human Reproduction 30(1):20-7. doi: 10.1093/humrep/deu286), Sunde et al., 2016 (Human Reproduction, Vol.31 (10): 2174–2182. doi:10.1093/humrep/dew157), Wale and Gardner 2016 (Human Reproduction Update 22(1):2-22. doi: 10.1093/humupd/dmv034), Simopoulou et al., 2018 (In Vivo 32(3):451-460. doi: 10.21873/invivo.11261).
  2. In table 1, replace Auto-fluoresence by Auto-fluoresCence; and Ultramicrofluoresence (UMF) by UltramicrofluoresCence (UMF)
  3. In Table 1, please add the references of the studies where the techniques mentioned have been used. This would help the readers to extract valid information.

  1. Ln 156: AUC refers to Area Under the Curve?

  1. In figure 1, replace long noncoding RNAs by long non-coding RNAs

Author Response

Some points to be expanded and corrected:

As for the reported heterogenous concordance rates between SBM and trophectoderm biopsy authors should discuss about the potential impact, if any, of different culture media and protocols among clinics. Also, for SBM technique, the different commercial culture medium, whose composition is unknown for the clinicians, can mask, increase or decrease the embryo secretion of specific biomarkers and metabolites. Same applies on the effect of culture media employed on morphological characteristics and embryokinetics. Some suggested papers to introduce this concept might be

Dyrlund et al. 2014 (Human Reproduction Nov;29(11):2421-30. doi: 10.1093/humrep/deu220),

 De Vos et al. 2015 (Human Reproduction 30(1):20-7. doi: 10.1093/humrep/deu286),

Sunde et al., 2016 (Human Reproduction, Vol.31 (10): 2174–2182. doi:10.1093/humrep/dew157),

Wale and Gardner 2016 (Human Reproduction Update 22(1):2-22. doi: 10.1093/humupd/dmv034),

Simopoulou et al., 2018 (In Vivo 32(3):451-460. doi: 10.21873/invivo.11261).

  • A new paragraph for the consideration of media impact has been added in section 3.

In table 1, replace Auto-fluoresence by Auto-fluoresCence; and Ultramicrofluoresence (UMF) by UltramicrofluoresCence (UMF)

  •  

In Table 1, please add the references of the studies where the techniques mentioned have been used. This would help the readers to extract valid information.

  • The table has been filled with relevant references, for the readers to easily extract valuable information.

Ln 156: AUC refers to Area Under the Curve?

  • Yes, the abbreviation has been added.

 In figure 1, replace long noncoding RNAs by long non-coding RNAs

  • corrected.

Reviewer 2 Report

Generally a nice concise mini review of the methods and techniques that have been studied with regards to using the secretome to assess embryo quality following IVF.

I only have some minor issues with the manuscript which are easily rectified:

Page 3 Table 1 - Immunohistochemistry should be removed; it's impossible to do this technique that uses tissue slices in spent media. Also not all of these spent media assays are discussed in the paper or mentioned at all.

Page 4 section 2.2 - would be useful to put AUC into abbreviations list.

Page 5 section 2.3 - 'The secretion of sCD146, as measured by immunohistochemistry, was shown to be associated with a lower implantation potential [32]'. This statement is wrong the authors of ref 32 declare they used immunocytochemistry on fixed embryos, and ELISA and Western blotting in spent media to look at levels of sCD146.

Page 6 section 3.1 - First use of MALDI-TOF MS, please put full name (currently in section 3.3).

Page 6 section 3.2 - To the sentence 'Twelve metabolites were analysed by 1H-NMR....' please add 'identified and' between were and analysed. Also add 1H-NMR to abbreviation list as proton-NMR, as not all readers will be familiar with the different types.

Page 7 section 3.3 - The term 'spectral fingerprinting' is not right in this context and will be mixed up with the technique of spectral fingerprinting which is analysing samples using different wavelengths of light and specifically requires no separation. MS fingerprinting would be a more appropriate term for what is described in ref 35.

Page 8 section 3.3 - There is much on the positivity of using MS in IVF clinics but nothing regarding the additional costs of having to employ specialists to run and maintain equipment and analyse data. Consider adding in.

Page 8 section 4 - MS-LC/GS-MS abbreviations are more commonly abbreviated to LC-MS and GC-MS. Consider changing to avoid confusion of the reader. Used LC/GS-MS in section 3.2 so consistency is needed too.

Page 9 abbreviations - see previous statement about MS-LC/GS-MS. MS is also missing from the end of the abbreviation explanation.

References - consistency is needed, some have all authors and some have first author and et al.. Preference would be for all authors to be listed. Reference 35 should have year changed to 2021 and doi added.

Author Response

I only have some minor issues with the manuscript which are easily rectified:

Page 3 Table 1 - Immunohistochemistry should be removed; it's impossible to do this technique that uses tissue slices in spent media. Also not all of these spent media assays are discussed in the paper or mentioned at all.

  • Immunohistochemistry was removed. The table has been filled with relevant references, for the readers to easily extract valuable information, thus it will be left even though not all techniques were discussed.

Page 4 section 2.2 - would be useful to put AUC into abbreviations list.

  •  

Page 5 section 2.3 - 'The secretion of sCD146, as measured by immunohistochemistry, was shown to be associated with a lower implantation potential [32]'. This statement is wrong the authors of ref 32 declare they used immunocytochemistry on fixed embryos, and ELISA and Western blotting in spent media to look at levels of sCD146.

  • Corrected in the text.

Page 6 section 3.1 - First use of MALDI-TOF MS, please put full name (currently in section 3.3).

  •  

Page 6 section 3.2 - To the sentence 'Twelve metabolites were analysed by 1H-NMR....' please add 'identified and' between were and analysed. Also add 1H-NMR to abbreviation list as proton-NMR, as not all readers will be familiar with the different types.

  • Corrected

Page 7 section 3.3 - The term 'spectral fingerprinting' is not right in this context and will be mixed up with the technique of spectral fingerprinting which is analysing samples using different wavelengths of light and specifically requires no separation. MS fingerprinting would be a more appropriate term for what is described in ref 35.

  • Corrected 'spectral fingerprinting' to 'mass spectral fingerprinting'.

Page 8 section 3.3 - There is much on the positivity of using MS in IVF clinics but nothing regarding the additional costs of having to employ specialists to run and maintain equipment and analyse data. Consider adding in.

  • I have added in the section on the cost and maintenance.

Page 8 section 4 - MS-LC/GS-MS abbreviations are more commonly abbreviated to LC-MS and GC-MS. Consider changing to avoid confusion of the reader. Used LC/GS-MS in section 3.2 so consistency is needed too.

  • Changed all instances to LC-MS/GS-MS

Page 9 abbreviations - see previous statement about MS-LC/GS-MS. MS is also missing from the end of the abbreviation explanation.

  • Corrected

References - consistency is needed, some have all authors and some have first author and et al.. Preference would be for all authors to be listed. Reference 35 should have year changed to 2021 and doi added.

  • All reference edited to be consistent. Ref 35 (currently 55), doi added, however it has appeared or available online: 09 December 2020, thus will leave it 2020.
